# Curcugreen Treatment Prevented Splenomegaly and Other Peripheral Organ Abnormalities in 3xTg and 5xFAD Mouse Models of Alzheimer’s Disease

**DOI:** 10.3390/antiox10060899

**Published:** 2021-06-02

**Authors:** Jayeeta Manna, Gary L. Dunbar, Panchanan Maiti

**Affiliations:** 1Field Neurosciences Institute, Ascension St. Mary’s Hospital, Saginaw, MI 48604, USA; jayeeta.manna@ascension.org; 2Field Neurosciences Institute Laboratory for Restorative Neurology, Central Michigan University, Mount Pleasant, MI 48859, USA; 3Program in Neuroscience, Central Michigan University, Mount Pleasant, MI 48859, USA; 4Department of Psychology, Central Michigan University, Mount Pleasant, MI 48859, USA; 5College of Health and Human Services, Saginaw Valley State University, Saginaw, MI 48604, USA

**Keywords:** Alzheimer’s disease, splenomegaly, antioxidant, curcumin, curcuminoids, inflammation, cytoprotection

## Abstract

Metabolic dysfunction and immune disorders are common in Alzheimer’s disease (AD). The mechanistic details of these epiphenomena in AD are unclear. Here, we have investigated whether a highly bioavailable curcuminoid formulation, curcugreen (CGR), can prevent abnormalities in peripheral organs of two mouse models of AD. Eighteen- and 24-month-old male and female 3xTg and 5xFAD mice were treated with CGR (100 mg/kg) for 2 months, orally. Cytoarchitectural changes of spleen, liver, kidney and lungs were studied by H&E stain. Apoptotic death was confirmed by TUNEL staining. Amyloid deposition, pTau levels, proinflammatory, anti-inflammatory and cell death/survival markers were studied by Western blots. Curcugreen reduced the observed splenomegaly (3xTg) and degeneration of spleen, granulomatous inflammation in the kidney, hepatic sinusoidal disorganization, hepatocellular hypertrophy, inflammation of the central hepatic vein, infiltration and swelling of lung tissues, and apoptotic death in all these areas in both 3xTg and 5xFAD mice. Similarly, CGR decreased amyloid deposition, pTau, proinflammatory markers, cell loss and decrements in anti-inflammatory markers in both 3xTg and 5xFAD mice. Peripheral organ abnormalities and inflammatory responses in AD were ameliorated by curcuminoid treatment.

## 1. Introduction

Although Alzheimer’s disease (AD) is considered as a central nervous system (CNS) disorder, accumulated epidemiological and experimental data demonstrated that there are systemic and/or peripheral inflammatory—along with innate—immune responses in patients suffering from AD [1,2,3,4]. Metabolic activities in the liver, kidneys and other accessory peripheral organs determine the state of physiological outcomes and maintain cardiovascular efficiency, whereas, decreased efficiency of peripheral organs leads to metabolic dysfunction which has a strong association with vascular injury and dementia [1,5]. 

Recent clinical studies have indicated that several metabolic disorders, such as diabetes, hypertension, obesity, dyslipidemia, and insulin resistance are the risk factors for AD, which are directly, or indirectly, associated with the impairment of energy metabolism, increased inflammation, and insulin resistance [6,7]. For example, recent evidence suggests that metabolic dysfunction of the liver, including significantly increased ratio of aspartate aminotransferase (AST) to alanine aminotransferase (ALT) levels, are associated with mild-cognitive impairment (MCI), AD, and other neurodegenerative changes [8]. Another study revealed that an increase in albumin concentration, total bilirubin, and lower levels of ALT were associated with increased Aβ deposition, reduced brain glucose metabolism, and greater brain atrophy, along with the cognitive deficits in AD [9]. Similarly, kidney dysfunction, especially reduced glomerular filtration rate (GFR), has been linked to the risk of dementia in community-dwelling older adults in China [10]. Further, individuals with abnormal lung function or chronic lung diseases have an increased risk of cognitive decline [11] and middle-aged adults with lung diseases may be at greater risk of developing dementia or cognitive impairment in their later life [12].

Exploring mechanistic details of peripheral organ dysfunction in AD could be facilitated by studying animal models of AD. The triple-transgenic mouse model of AD (3xTg-AD or simple 3xTg) is a well-established animal model, due to development of both Aβ plaques and neurofibrillary tangles, which are the hallmark pathologies typically observed in patients with AD [13]. In this model, three human genes coding for amyloid precursor protein (APPswe), presenilin 1 (M146V) and human tau (TauP301L) are modified, which induces synaptic dysfunction and cognitive impairment [14,15,16]. Similarly, 5xFAD is another well-established mouse model of AD, where five human genes are genetically modified, including three mutations on amyloid precursor protein (APP) genes, such as mutations in Swedish-(K670N, M671L), Florida- (I716V), as well as and London (V717I) and two mutations on PS1 genes, such as M146L and L286V [17]. These five mutations induce progressive deposition of Aβ plaques in the selected regions of the brain, along with concomitant behavioral deficits, including memory loss and cognitive impairments at about 2 months of age [18] and it is progressive with age. 

Previous reports demonstrated that the 3xTg mice can have severe splenomegaly, along with inflammatory responses, alteration of plasma cytokines and antibodies formation, along with behavioral deficits [19]. Some reports suggested that CD4+ and CD8+ T-cells were significantly altered in the spleen of 3xTg mice, along with increased levels of interleukin 6 (IL-6), IL-1β and C-reactive protein [20]. It is not clear how the spleen becomes extensively enlarged in 3xTg and how the immune system outside of the CNS is affected in AD patients. These observations depicted the importance of studying different peripheral metabolic organs, such as liver, kidney, lungs and immune organs for deciphering the pathophysiology of AD, which could help bridge the gap in understanding how the between central and peripheral systems might be interacting in AD. Taken together, we were interested in doing a systematic search for any significant alterations in peripheral organs and immune systems of two different, well-established AD mouse models and to determine whether we can prevent these alterations by treatments with anti-inflammatory natural compounds.

One natural compound, curcumin (Cur) has antioxidant, anti-inflammatory, and anti-amyloid properties, and has been extensively investigated, including thorough work in our own laboratory, as a potential treatment for AD [21,22,23,24,25,26,27,28,29]. The most active compound in turmeric (a root of the herb *Curcuma longa)* is Cur and other important compounds are bismisthoxycurcumin (BMC) and debismethoxycurcumin (DBMC). These three compounds, together, are known as curcuminoids, can cross the blood brain barrier (BBB), penetrate into brain cells, and protect neurons from Aβ-induced neuronal death [25,29]. These compounds are hydrophobic or lipophilic in nature, but possess poor solubility in most body fluids, limiting their bioavailability. Therefore, many investigators have developed different lipids or nano-conjugated formulations to increase their bioavailability. For example, solid lipid curcumin particles (SLCP) have been shown to prevent neuropathology in AD and other neurodegenerative diseases [25,26,29,30]. More recently, we have been investigating the effects of curcugreen (CGR) on animal models of AD. The CGR contains about 86% curcuminoids with 7–9% ar-turmerone, an essential oil from turmeric. When taken orally, CGR provides the free Cur which is retained in the blood for a longer period and has 700% more potency than regular turmeric [31].

The objective of this study was to investigate the overall histological changes and cell death mechanisms in spleen, liver, kidney, and lungs of the 3xTg and 5xFAD mouse models of AD after treatments with CGR or vehicle for 2 months. 

## 2. Materials and Methods

### 2.1. Chemicals and Reagents 

All the chemicals, other accessories and antibodies used in this study are documented in Appendix A as a Appendix A. The CGR was gifted from Arjuna Natural Private Limited, Kerala, India. The detailed bioavailability of CGR has been documented by Barret ML [31]. 

### 2.2. Animals

Male and female triple transgenic (3xTg, 18- and 24-month-old) and 5xFAD (24-month-old) mice and their respective age-matched wild-type mice were used for this study. The animals were bred in the vivarium at the College of Health and Human Services, Saginaw Valley State University. The 3xTg mice colony was developed by breeding of homozygous 3xTg-AD mice possessing PS1M146V, APPswe, and *tau*P301L transgenes with control, wild type (WT) mice procured from Jackson laboratory (Jackson Laboratories, Bar Harbor, ME, USA) at the 8 weeks of age. Therefore, heterozygous 3xTg mice were used for this study. Similarly, for 5xFAD mice, the male and female mice overexpressed human APP and PS1 with five familial AD mutations, including three mutations on APP gene [Swedish (K670N, M671L), Florida (I716V), and London (V717I)] and two on PS1 gene (M146L and L286V). Detailed pathological and behavioral phenotypes were described by many investigators previously. All mice were genotyped at 3 weeks of age by polymerase chain reaction (PCR) to confirm their transgenic characteristics. Details of experimental groups, and treatment regimen were documented in Appendix A in the Appendix A. 

All mice were housed (4 mice/cage) and kept under standard laboratory conditions: light phase from 600 to 1800 h, room temperature of ~25 °C, humidity of ~66%, and low fat rodent chow and tap water provided ad libitum. All protocols were performed in accordance with the rules and regulations of the Institutional Animal Care and use Committee at Saginaw Valley State University (protocol no: 1140960-2). All procedures were performed under proper anesthetic conditions, and all efforts were made to minimize animal discomfort.

### 2.3. Curcugreen Solubility and Treatment

Curcugreen (CGR), also known as BCM-95^®^ was developed and patented by Arjuna Natural Pvt. Ltd. (Arjuna natural, Kerala, India). It contains 86% pure curcuminoid extract of turmeric with 7–9% ar-turmerone oil. It is highly bioavailable (700% more potency than standard turmeric 95% extract). The free curcuminoids retained in the bloodstream even after 8 h of oral intake. The CGR was dissolved in 0.5% methylcellulose in PBS (0.1 M, pH 7.4) and stored at 4 °C. The CGR solution was administered to the mice, via oral gavage, for 2 months, every other day, at the dose of 100 mg/kg body weight, while age-matched WT mice received equivalent amounts of vehicle (0.5% methylcellulose in PBS).

### 2.4. Tissue Collection 

For histological studies, the mice were deeply anesthetized with an overdose of Fatal-Plus (0.22 mL/kg of body weight, i.p.) and blood was collected from the hepatic vein. Then the mice were transcardially perfused with 0.1 M cold PBS at pH 7.4, followed by a 4% paraformaldehyde (diluted in 0.1 M PBS at pH 7.4) fixation solution. The liver, kidney, spleen and lungs were dissected out and post-fixed in 4% paraformaldehyde and stored at 4 °C until use. Whereas for Cur measurement, the mice were sacrificed by cervical dislocation and then the liver, kidney, spleen and lungs were collected and stored at −80 °C until they were processed for ultra-performance liquid chromatography (UPLC).

### 2.5. Curcumin Measurement by Ultra Performance Liquid Chromatography (UPLC)

About 100 mg of flash frozen liver, kidney, cerebellar tissue and blood (~0.5–1 mL) were taken in a 1.5 mL Eppendorf tube. The frozen tissue was homogenized by a hand homogenizer in ice-cold condition with 10 volume of 1 M ammonium acetate (pH 4.6) and then centrifuged at 13,000 rpm for 10 min at 4 °C. The supernatant was taken and diluted with 2 volumes of extraction buffer (95% ethyl acetate and 5% methanol). The solution was mixed by vortex and then centrifuged for 10 min at 3000 rpm. The supernatant was placed into another glass tube and the extraction process was repeated. Then the supernatant was dried under gentle nitrogen flow in a laminar hood. The dried materials were dissolved in solvent B (acetonitrile: water with 0.15% formic acid, 50:50) and then the mixture was filtered with 0.2 μm filter. Ten-μL filtrate was injected to UPLC and the peak area was measured, and the free Cur level was calculated by using standard curve. The standard curve for Cur was made by using 1, 5, 10, 25, 50 and 100 μM of Cur solution dissolved in solution B and injected to UPLC and the area of the Cur peaks were measured to achieve standard curve. 

### 2.6. Tissue Processing for Histology

About 2–3 mm tissue was trimmed and processed for paraffin embedding. Briefly, the tissue was dehydrated with 70%, 80% alcohol, followed by 95% ethanol and 5% methanol for 1 h each. Then three changes in absolute alcohol, 1 h each, followed by two changes in xylene (45 min each). The tissue was embedded with paraffin at 56 °C for two changes, 1 h each, followed by another 4 h embedding with fresh paraffin solution. The paraffin block was made and a 5 μm thick section was cut using a rotary microtome (Microm, HM-310, Charleston, SC, USA) and placed on a poly-L-lysine-coated glass slide. 

#### 2.6.1. Hematoxylin and Eosin (H&E) Stain

The paraffin-embedded sections were stained with hematoxylin-eosin (H&E). Briefly, the sections were deparaffinized by dipping in xylene, two times, 5 min each, followed by immersion in consecutive 100%, 90%, 70% and 50% solution of alcohol for 1 min each and then immersed in tap water for another 5–10 min. Then the sections were stained with Mayer’s hematoxylin solution for 5 min, followed by an immersion in tap water for 10 min. The tissue was treated with a bluing solution for 1 min, followed by a wash in tap water for another 5 min. Then the tissue was decolorized with 1% acid alcohol (1:1) and dipped in tap water for another 5 min. The tissue was then treated with 70% alcohol and counter-stained with eosin-Y for 2 min, and washed with 70% alcohol for two times, 1 min each. Then the sections were dehydrated with graded alcohol and cleared with xylene (5 min each, two times) and mounted using DePex mounting media. Images were taken using a compound light microscope (Olympus, Japan) using 10x and 40x objectives. 

#### 2.6.2. Terminal Deoxyribonucleic Acid Nick End Labeling (TUNEL)

DNA fragmented cells were labeled by TUNEL Assay Kit-BrdU-Red as described, previously [32,33]. Briefly, paraffin-embedded liver, kidney, spleen and lung tissues were sectioned at 5 μm thickness using rotary microtome and taken on a poly-L-lysine-coated glass slides. The sections were deparaffinized and rehydrated and then treated with 0.85% sodium chloride solution for 5 min, followed by three washes with PBS (0.1 M, pH 7.4), 5 min each. Then the sections were treated with proteinase K (20 ng/μL) for 5 min, followed by three washes with PBS, 5 min each. The sections were then incubated with BrdU solution for 1 h at 37 °C in a humidified chamber in the dark, followed by washing three times with PBS, 5 min each. Then the sections were incubated with anti-BrdU antibody conjugated with Texas-red, followed by three washes with PBS, 5 min each. All sections were counterstained with DAPI (20 mmol/L) for 10 min at room temperature in the dark and washed thoroughly with distilled water before being mounted with an anti-fading medium. The images were captured using fluorescent microscopes (Leica, Germany) by 40x objectives (total magnification 400×) with appropriate excitation/emission filters (ex/em: 488/576), so that the TUNEL-positive cells produced red-fluorescence. The number of TUNEL-positive cells were counted manually in each microscopic image and expressed as the number of TUNEL positive cell/microscopic fields. At least 30–40 images were taken from three mice of each group to obtain a mean value. The counting was performed independently by two researchers, who were blind to the group identity of the sample.

#### 2.6.3. Labeling of Amyloid Deposition with Congo Red (CR) Staining

The CR staining was performed to label the amyloid deposits in the peripheral organs, especially in the spleen and in the liver with some modifications [23]. Briefly, the paraffin embedded sections (5 μm thick) were deparaffinized, and rehydrated with graded alcohol solution and then washed with lukewarm tap water for 20–30 min. The sections were stained with 0.5% CR solution, dissolved in 80% ethanol along with 1% NaOH for 1 h at room temperature. Then the sections were rinsed in running, lukewarm tap water for 15–20 min, followed by counterstaining with Mayer’s hematoxylin solution for 5 min and washed three times with distilled water, dehydrated through an ascending alcohol series (70%, 90%, and 100%), and then placed into xylene and cover-slipped with DePeX. The sections were visualized using a compound light microscope (Olympus, Japan) using 40× objectives (total magnification = 400×) with a radish pink color indicating amyloid deposition. 

#### 2.6.4. DAPI and Propidium Iodide Staining

For DAPI and PI staining, the paraffin-embedded sections were deparaffinized and rehydrated with a graded alcohol series (70%, 90%, and 100%) and then stained with DAPI (20 mmol/L) for 10 min at room temperature in the dark. For propidium iodide (PI) staining, the sections were stained with 500 nM concentration of PI for 5 min at room temperature, in dark, after which the sections were washed with distilled water for tree time, 5 min each, and mounted with aqueous anti-fading mounting media. The images were captured by a fluorescent microscope (Leica, Wetzlar, Germany) using the appropriate excitation/emission filters. The fluorescent intensity of DAPI and PI stained images were measured manually using ImageJ software and expressed as fluorescent intensity in arbitrary units (AU). 

#### 2.6.5. Immunohistochemistry of Phosphorylated Tau

The immunohistochemistry protocol for pTau was described elsewhere [23,24,25,26,27,28]. Briefly, the paraffin sections were first deparaffinized and rehydrated with graded alcohol. Then the sections were washed with PBS-Triton-X-100 (0.5%, PBS-T) for 5 min and blocked with 10% goat serum (dissolved in PBS-T) for 30 min at room temperature with gentle shaking. The sections were incubated with pTau antibody (mouse monoclonal, 1:200), overnight on the shaker. On the next day, the sections were washed three times, 5 min each, and incubated with anti-mouse secondary antibody, conjugated with Alexa-488, for 1 h at room temperature with gentle shaking. Then the sections were washed with water, three times, 5 min each, followed by counterstaining with PI (500 nM) for 5 min. Finally, the sections were washed, and mounted with anti-fading mounting media and the images were taken using fluorescent microscopes (Leica, Wetzlar, Germany) with appropriate excitation and emission filters. The bright green signal was considered as pTau-positive cells. The number of pTau-positive cells were counted manually from the spleen tissue of three mice in each group with at least 15–20 randomly captured images and expressed as mean pTau positive cells/microscopic field.

#### 2.6.6. Immunoperoxidase Staining for Cleaved Caspase-3

The detailed protocol for immunoperoxidase techniques for caspase-3 was followed from the previously described protocol with some modifications [25]. Briefly, the paraffin sections of liver, kidney, spleen and lungs were deparaffinized, and rehydrated. The sections were then incubated with 3% hydrogen peroxide solution for 30 min at room temperature to inhibit endogenous peroxidase. Then the sections were rinsed with PBS, three times, 5 min each, and blocked with 10% normal goat serum (NGS), and underwent gentle shaking for 1 h at room temperature. The tissue sections were then incubated with cleaved caspase-3 antibody (rabbit, monoclonal, 1:200) with gentle shaking overnight at 4 °C. On the next day, the sections were thoroughly washed 3 times, for 5 min each, and incubated with a biotinylated anti-rabbit secondary antibody (1:250) for 1 h at room temperature. Finally, the sections were incubated with a peroxidase substrate solution, supplied with the ABC kit (Vector Laboratory, Burlingame, CA, USA), and the signal was developed using diaminobenzidine (DAB) until the desired staining intensity emerged. The sections were counterstained with hematoxylin for 10 min and then washed, dehydrated with graded alcohol, cleared with xylene, mounted on slides, cover slipped with DePex, air dried, and visualized using a compound light microscope (Olympus, Japan) with 40x objectives (total magnification 200×). 

#### 2.6.7. Western Blot

Detailed protocol for Western blot techniques was described previously [24]. Briefly, about 100 mg of flash-frozen spleen tissue from WT, 3xTg, 5xFAD and the 3xTg or 5xFAD mice treated with CGR were lysed with cold radioimmunoprecipitation assay (RIPA) buffer with protease and phosphatase inhibitors. The total protein was measured by the bicinchoninic acid assay (BCA assay) and about 200 μg of protein from each group was loaded on 4–20% Tris-glycine gel and separated by sodium dodecyl sulphate polyacrylamide gel electrophoresis (SDS-PAGE). The proteins were transferred to a polyvinylidene fluoride (PVDF) membranes and the membranes were probed with different primary antibodies (1:1000), such as caspase-3, 6, Bcl_2_, TNF-α, IL-1β, IL-10, pTau, pAkt and pGSK-3β followed by their respective secondary antibodies (1:10,000). The blots were developed with SuperSignal™ West Femto Maximum Sensitivity Substrate using gel documentation system (Bio-rad). The relative optical density was measured using Image-J software (https://imagej.nih.gov/ij/) (accessed on 2 April 2021). To ensure equal protein loading in each lane, the blots were re-probed for GAPDH antibodies.

#### 2.6.8. Statistical Analysis

The data were analyzed using the one-way analysis of variance (ANOVA), followed by Tukey’s honestly significant difference (HSD) post hoc test. All the data were expressed as mean ± SEM and the *p* ≤ 0.05 was considered as significant. 

## 3. Results

### 3.1. Curucmin Levels in Different Tissues in 3xTg and 5xFAD Mice after Treatment with CGR

To compare the solubility of CGR in water, we dissolved equal amounts of natural curcuminoids and CGR in a similar volume of distilled water and observed that the natural curcuminoids were precipitated, whereas CGR completely dissolved in distilled water (Figure 1A). Given this, we measured the bioavailability of free Cur in different peripheral organs, including blood after treatment with CGR (100 mg/kg), using UPLC from these tissues. We found ~4–5 μM of free Cur in 3xTg and ~12–14 μM in the blood of 5xFAD mice (Figure 1D), whereas ~2.5 μM in 3xTg and ~1.8 μM in the cerebellum of 5xFAD mice (Figure 1E). Further, we found ~1 μM and ~2 μM of free Cur in the liver and kidney, respectively in 3xTg mice and ~6 μM and ~4.5 μM of free Cur in the livers and kidneys, respectively in 5xFAD mice (Figure 1F–G). 

### 3.2. Splenomegaly in 3xTg Mice Was Prevented by CGR Treatment

One of the aims and objectives for this study was to investigate and compare the degree of splenomegaly in 18-month-old and 24-month-old 3xTg and in 18-month-old 5xFAD mice and whether the treatment of CGR has any effect. We distinctly observed an extensively enlarged spleen in both genders of the 18- and 24-month old 3xTg mice. The 24-month group showed a greater degree of change in spleen size when compared to 18-month old 3xTg mice (Figure 2B,E). Treatment with CGR significantly decreased the spleen size in both 18- and 24-month old groups of 3xTg mice (Figure 2C,F). No significant differences were observed in spleen size in 5xFAD mice, compared to their age-matched WT mice (Figure 2G–I). 

### 3.3. Morphological Damage of Splenic White- and Red-Pulp Were Prevented by CGR Treatment in Both 3xTg and 5xFAD Mice

We performed H&E stains to investigate the histological changes in both 3xTg and 5xFAD mice. There were structural abnormalities, including the destruction of splenic microarchitecture with significantly decreased lymphocytic loss in both the white-pulp (Figure 3) and red-pulp (Figure 4). The lymphocytes in the red-pulp disappeared accompanied by the appearance of polynuclear neutrophils and monocytes in both 3xTg and 5xFAD mice (Figure 4). When we stained the white pulp and red pulp with DAPI and PI and then measured the fluorescent intensity (arbitrary units), there was a significant decrease in fluorescent intensity in both white- and red-pulp (Appendix A) of spleen in the 3xTg mice and CGR treatment appeared to normalize this similar to what was observed in WT mice (Figure 3 and Figure 4 and Appendix A). 

### 3.4. Morphology of the Liver Was Restored by CGR Treatment in Both 3xTg and 5xFAD Mice

A significant degeneration and infiltration of the central hepatic vein, hepatocellular hypertrophy, and brown pigmentation were observed in both 18- and 24-month-old 3xTg mice (Figure 5A,B). Similarly, a disorganization of central vein, sinusoids, and hepatocellular hypertrophy were also observed in the 5xFAD mice (Figure 5C and Appendix A). Treatment with CGR restored the liver cytoarchitecture in both the animal groups. 

### 3.5. Kidney Morphology Was Restored by CGR Treatment in 3xTg and 5xFAD Mice

We were interested to investigate if there is any effect on kidney tissue in these AD mice and whether CGR treatment could ameliorate these effects. We observed a significant shrinkage and degeneration of Bowman’s capsule and glomerulus in both the 3xTg (Figure 6A,B) and 5xFAD mice (Figure 6C). Infiltration of blood and swelling or increase in Bowman’s space, decrease in filtered cells, such as podocytes in the glomerular capsule has been observed in the 3xTg mice. A significant amount of granulomatous inflammation was also observed in the 5xFAD mice (Figure 6C and Appendix A) which were prevented by CGR treatment.

### 3.6. Lung Morphology Was Restored by CGR Treatment in Both the 3xTg and 5xFAD Mice

Lung morphology was studied by H&E stain, as well as with DAPI and PI stains (Appendix A). We observed a rupture of blood vessels, and infiltration of blood in the alveolar spaces, along with bronchiolar degeneration in both 3xTg (Figure 7A,B) and 5xFAD mice (Figure 7C). Treatment with CGR prevented these changes in both the groups. 

### 3.7. Apoptotic Death (DNA Fragmentation) Was Prevented by CGR Treatment in the Spleen, Liver, Kidney and Lung Tissue of 3xTg and 5xFAD Mice

One of the objectives of this study was to assess the mode of cell death in peripheral organs of 3xTg and 5xFAD mice. We performed TUNEL staining to evaluate the DNA fragmented (apoptotic) cells in both white- and red-pulp of spleen and in the liver, kidney and lung. We observed a significant increase in the number of TUNEL-positive cells in the spleen (Figure 8A–C), liver, kidney and lungs (Figure 9A,B) in both 3xTg and 5xFAD mice and treatment with CGR significantly prevented the number of TUNEL-positive cells in these areas. More TUNEL-positive cells were noted in the case of 3xTg mice in comparison to 5xFAD mice and the spleen showed a greater number of TUNEL-positive cells than in other peripheral organs. 

### 3.8. Amyloid Deposits in the Peripheral Organs in the 3xTg and 5xFAD Mice Were Reduced by CGR Treatment

The amyloid deposits were stained with Congo-red stain, which binds with amyloid fibrils of several proteins. We observed numerous pink-red deposits of amyloid (arrow) in all the tissues sampled in both 3xTg and 5xFAD mice. We found more amyloid deposits in the liver and the least amount was observed in the kidney tissue. When we counted the number of amyloid positive cells in the liver and spleen tissue, we found a significant increase in number of amyloid positive (CR positive) cells in both these areas in comparison to vehicle-treated 3xTg and 5xFAD mice, whereas treatment with CGR significantly reduced the CR-stained amyloid deposition in these tissues (Figure 10C,D). Similar phenomena were also observed in the case of lungs and kidney tissues (graphs not shown). 

### 3.9. Phosphorylated Tau Immunoreactivity Was Reduced by CGR Treatment in 3xTg Mice

The 3xTg mouse has the human transgene for microtubule-associated protein tau (MAPT) which produces phosphorylated tau (pTau) in the brain tissue. We were interested in whether 3xTg mice produce any pTau in the peripheral organs and whether CGR treatment ameliorated their levels. Therefore, we performed immunohistochemistry of pTau in the liver, kidney, spleen and lungs. We observed a significantly increased number of pTau-immunoreactive (pTau-IR) cells in all these organs, especially in the spleen (Figure 11). Interestingly, CGR treatment completely prevented accumulation in these organs (Figure 11B). Similarly, our Western-blot data showed that there was an increase in pTau (Figure 11C,D) and tau kinase, such as pGSK-3β in 3xTg mice (Figure 11C,E) and CGR treatment significantly reduced these levels.

### 3.10. Cleaved Caspase-3 Immunoreactivity in 3xTg and 5xFAD Mice after Treatment with CGR

To correlate the TUNEL staining data, we performed the immunohistochemistry of cleaved caspase-3 in the spleen, liver, kidney and lung tissue of 3xTg and 5xFAD mice after treatment with CGR. We observed an increased in immunoreactivity of cleaved caspase-3 in the white- and red-pulp of the spleen, liver, kidney and lung in both 3xTg (Figure 12A) and 5xFAD tissue (Figure 12B) and treatment with CGR partially reduced this immunoreactivity. We found a greater immunoreactivity in the case of 3xTg than 5xFAD mice (Figure 12A,B). In addition, the spleen tissue showed greater caspase-3 immunoreactivity in comparison to the other three tissues studied in both these AD mice models.

### 3.11. Cell-Death and Cell-Survival, Inflammatory and Anti-Inflammatory Markers Were Normalized by CGR Treatment in the Spleen of 3xTg and 5xFAD Mice

To investigate some of the cell death, cell-survival, inflammatory, and anti-inflammatory-markers from the spleen tissue of 3xTg and 5xFAD mice, we performed Western blot. We observed a significant increase in levels of caspases 3, 6, TNF-α, IL-1β and a significant decrease in Bcl_2_, IL-10, COX-IV and pAkt/Akt levels in both the 3xTg and the 5xFAD mice (Figure 13), whereas, treatment with CGR restored their levels. 

## 4. Discussion

Although the central nervous system is the most vulnerable area affected by Alzheimer’s disease (AD), recent evidence indicates that AD is a multi-organ disorder, which includes metabolic dysfunction and pathological processes in the peripheral organs [30]. Therefore, we were interested in investigating changes in the peripheral organs in two animal models of AD. We compared the morphological changes of liver, kidney, spleen, and lungs in the 3xTg mice and 5xFAD mice, which are well-established animal models of AD. In addition, we tested whether curcuminoids, which are natural polyphenols with an anti-amyloid, and anti-inflammatory properties can ameliorate these peripheral pathological changes. We observed a splenomegaly in the 3xTg mice, but not in the 5xFAD mice. There were significant cytoarchitectural changes of the liver, kidney, spleen and lungs in both 3xTg and 5xFAD mice, along with increased apoptosis in these organs, for which curcugreen (CGR), a highly bioavailable curcumin formulation was able to prevent these morphological alterations and apoptotic cell death in both the 3xTg and 5xFAD mice.

Previous studies have indicated that 3xTg mice develop splenomegaly and autoimmune responses [19], but the molecular mechanisms for these changes are unclear. Interestingly the 5xFAD mouse model of AD, which has both APP and presenilin-1 gene modifications does not display splenomegaly. Therefore, it is speculated that the 3xTg mouse models of AD which contain the human tau gene may have some role in the development of autoimmune responses and splenomegaly. In addition, if the immune system is dysregulated in AD, then there might be a strong association of spleen changes along with other peripheral metabolic organs, such as liver, kidney or lung. Given that if severe manifestations of systemic autoimmune or inflammatory responses are the cause for enlargement of the spleen, then anti-inflammatory natural compounds, such as curcuminoids could be effective to ameliorate these changes. All this unresolved information prompted us to explore the mechanisms of these epiphenomena in 3xTg and 5xFAD mice, because both these mouse models have been extensively used because of their ability to mimic the pathophysiology observed in the human AD patients.

As immune reaction is prominent in AD mice, we were interested in reducing the immune reaction by anti-inflammatory natural compounds called curcuminoids, which include Cur, bisdemethoxycurcumin (BDMC), and demethoxycurcumin (DMC). We used CGR which contains 86% of curcuminoids, and these compounds have been shown to have strong anti-inflammatory properties, as reported by many investigators [21,22], including the results from our own laboratory [23,24,25,26,27,28,29]. There are certain advantages to use CGR for our study: (i) it is more water soluble (Figure 1A); (ii) it is 700% more potent than natural turmeric; and (iii) it can stay for a longer period (8–10 h) in the systemic circulation in comparison to the natural curcuminoids, when administered orally. As Cur is the most active compound in turmeric [34,35], we were interested to know how much free Cur was available in different tissues of mice treated with CGR for 2 months. Therefore, we measured free Cur levels from blood, liver, kidney and brain tissues from 3xTg and 5xFAD mice using ultra performance liquid chromatography (UPLC). In general, after oral administration of natural Cur, only 25–50 nM free Cur is available in the systemic circulation and becomes excreted or degraded within 2–4 h of its administration. However, we found approximately 4–13 μM free Cur in blood, 2–3 μM in cerebellar tissue, 1.5–6 μM in the liver, and 2–4 μM from the kidney tissue (Figure 1), which is 200–300 times more than natural Cur. Interestingly, we observed greater bioavailability of free Cur in most of these tissues in the 5xFAD mice in comparison to 3xTg mice (Figure 1D–G), although further research is needed to explain these differences. 

When we looked at the spleen structure, we observed multiple white patches, along with extensive splenomegaly in 50–60% of mice of either sex, at 18–24 months of age in 3xTg mice, as reported, previously [30]. We also found a greater splenomegaly in 24-month-old 3xTg mice in comparison to those 18-months of age (Figure 2A–F), including a greater mortality rate, which indicates that with age, the immune responses were enhanced, along with increased spleen size. In contrast, we did not observe any changes in overall size of the spleen in the case of 5xFAD mice (Figure 2G–I), which indicates that 3xTg, with the human tau gene, could have some role in increasing immune responses and splenomegaly. Interestingly, in the present study, CGR treatment significantly reduced the size of the spleen in both the 18- and 24-month-old 3xTg mice (Figure 2), which suggests that Cur has ameliorative roles in reducing spleen size. 

Abnormal spleen size or increase inflammatory responses are associated with dysfunction of other metabolic organs, therefore one of the aims and objectives of this study was to investigate the cytoarchitectural changes of different peripheral organs in both the AD mouse models and determine whether reduction of inflammatory responses by the CGR treatment could protect against these pathological changes. We performed hematoxylin and eosin (H&E) stains in the spleen, liver, kidney and lungs to check their cytoarchitectural changes. The splenic cytoarchitecture was severely affected (Figure 3 and Figure 4), as seen by a decreased number of lymphocytes, along with degeneration of the white- and red-pulp areas, disorganization of the splenic nodules and extracellular vacuolization, a finding in concert with the work of Yang and colleagues [33]. When we measured the fluorescent intensity of DAPI and PI from the white pulp area of spleen, we clearly observed a significant decrease in fluorescent intensity in 3xTg mice in comparison to its age-matched WT and 3xTg+CGR-treated mice (Appendix A), indicating a decrease in splenic lymphocytes in 3xTg untreated mice. Previous reports suggested that the number of CD4+ and CD8+ lymphocytes were significantly decreased in 3xTg [36], which strengthens our findings. In addition, more neutrophils and monocytes infiltrate in the red pulp area of the spleen in the 3xTg mice, indicating increased inflammatory responses, which was decreased by CGR treatment (Figure 4A,B).

Peripheral organs, such as the kidney and the liver, play an essential role in the clearance of circulating amyloid proteins [30,37]. Therefore, we have investigated the liver and kidney cytoarchitectural changes and we observed that these organs were also significantly affected in both 3xTg and 5xFAD mice. We found an increase in TUNEL-positive, DNA-fragmented cells (Figure 8 and Figure 9), along with disorganization of central vein, hepatocyte hypertrophy, and infiltration of blood in the sinusoid area, as well as around the central hepatic vein, as observed by H& E stains (Figure 5) and by DAPI and PI staining (Appendix A). It is speculated that these degenerative changes of liver might alter some vital metabolic enzymes in liver, such as aspartate transaminase (AST), alanine transaminase (ALT), alkaline phosphatase (ALP), as reported by Nho and colleagues [8,37]. In fact, decreases in ALT level is associated with disturbances of liver glucose metabolism, along with a greater Aβ deposition and structural atrophy in patients with AD, as reported previously [8]. These changes are associated with reduced brain function, including lower scores on measures of memory and executive function in AD patients. In addition, liver damage is also associated with cardiovascular disease, insulin resistance via disturbances of lipid metabolism, which indirectly affect excretory and respiratory systems [8]. In contrast, CGR treatments protect the liver cytoarchitecture by maintaining sinusoidal structure, and the integrity of the central vein. Although we did not measure any alteration of liver enzymes in 3xTg and 5xFAD mice, which are gold standards for assessing hepatic injury, we did observe CGR-treatment decrease in TUNEL-positive cells (Figure 8 and Figure 9), improvements in hepatic morphology (Figure 5 and Appendix A), decreases in hepatic white patches and hepatomegaly, which suggests that CGR treatment is hepatoprotective.

The 3xTg mice displayed splenomegaly, hepatomegaly, elevated levels of pro-inflammatory markers, such as TNF-α and decreased lymphocyte population in spleen, which indicates a severe manifestation of systemic autoimmune/inflammatory responses. Previous reports [33] suggested that “double-negative” T-cells produce pro-inflammatory cytokines, such as TNF-α levels, which support our findings. Importantly, lymphocytic populations were restored by CGR treatment in both 3xTg and 5xFAD mice, along with a decrease in the levels of TNF-α and IL-1β (Figure 13). In addition, a decrease in Bcl_2_, COX-IV and pAkt/total Akt levels (Figure 13), along with increased TNF-α and IL-1β levels, indicate that the cells were under stress or injury or damage, whereas CGR treatment significantly decreased TNF-α, IL-1β and increased cell-survival markers, such as pAkt/total Akt and Bcl2 indicating CGR protected cells from these immune reactions. For example, decreasing COX-IV (an important component of mitochondrial electron transport chain complex-IV) in 3xTg and 5xFAD mice, indicates the mitochondrial damage, which is associated with cellular injury, is protected by CGR treatments. We correlated the TUNEL staining data with cleaved caspase-3 immunoreactivity and Western blot for caspase-3 and 6, and we observed an increased cleaved caspase-3 in 3xTg mice, suggesting an increase in apoptotic death (Figure 12 and Figure 13). Although, we observed slightly different scenarios in the case of 5xFAD mice, whereby decreased levels of TNF-α in 5xFAD, in comparison to WT, were observed. We are unclear about the underlying causes of these discrepancies, but our data suggest that the inflammatory events in 3xTg might be different from those of 5xFAD mice.

Furthermore, we have investigated the anti-inflammatory cytokines, such as IL-10 which inhibit proinflammatory cytokines. Although we did not observe any significant change of IL-10 levels in 3xTg and 5xFAD mice in comparison to their age-matched WT mice, CGR treatment significantly increased their levels in both 3xTg and 5xFAD mice (Figure 13A,H,I), confirming the strong anti-inflammatory properties of CGR which might induce the levels of IL-10, along with other anti-inflammatory cytokines, as well as other cell survival markers. Additionally, we are aware that the enzyme-linked-immunosorbent assay (ELISA) might be a more appropriate to quantify the levels of proinflammatory and anti-inflammatory markers, our Western blot data clearly indicated a signal that CGR efficiently ameliorated the proinflammatory markers by inducing anti-inflammatory cytokines in these AD mouse models. 

As strong anti-inflammatory compounds, curcuminoids have been shown to protect against inflammation-induced tissue damage. Therefore, cytoprotective effects of Cur, including decreased TUNEL-positive cells and an increase in cell-survival markers, such as Blc_2_ and pAkt/Akt levels, further support the previously reported findings [21,36]. We know that the increased amyloidosis has a strong correlation with an increase in inflammatory responses in AD. Therefore, we thought reducing either pTau or Aβ levels could be a viable approach to attenuate the overall pathological changes in these organs. Previous studies revealed that Cur prevented tau dimerization and higher order oligomers formation in a human tau transgenic mouse model [21]. Therefore, we were interested in investigating and correlating the pTau levels and inflammation in the 3xTg and 5xFAD mouse models. We performed Congo-red staining and pTau immunohistochemistry (IHC) in all these sampled peripheral organs. Congo-red is a classical amyloid dye to label misfolded amyloid proteins in tissue, as reported by many investigators, including our laboratory [23]. We observed a significant amount of amyloid depositions (CR-positive cells) in all these peripheral organs, especially in the liver and spleen (Figure 10C,D) in both the 3xTg and 5xFAD mice (Figure 10). We observed greater amyloid levels in the liver (Figure 10A,C) and less amount was observed in the kidney in both these AD models (Figure 10). Increased amyloidosis correlated with degree of tissue damage and apoptotic death. Further, the amyloid deposition was also correlated with the phosphorylated tau (pTau) immunoreactivity levels in all these tissues, especially in the spleen of 3xTg mice (Figure 11). We found a decrease in pTau levels by CGR treatment in 3xTg mice (Figure 11), suggesting curcuminoids inhibited either pTau formation or might degrade their levels through other mechanisms, such as the proteasomal system. Decreasing pTau level by CGR treatment was supported by the levels of pGSK-3β which is considered one of the principal tau kinases (Figure 11C,E) to produce pTau in vivo. 

Decreasing pTau or inflammatory markers by CGR treatment in peripheral organs may not correlate with the brain tissue. In fact, the brain pathophysiology of 3xTg and 5xFAD mice have been well-documented by several researchers. Therefore, our focus in this study was to investigate the changes of peripheral organs by comparing 3xTg and 5xFAD mice and we did not investigate the amyloid and pTau levels or neuroinflammatory markers in brain tissue in this study after treatment of CGR. However, in recent studies, we have demonstrated neuroprotective and behavioral improvement in 5xFAD mice after treatment with lipid formulation of curcumin [28]. Additionally, we did not investigate the important metabolic enzymes from these peripheral organs, especially the serum glutamate oxaloacetate transaminase (SGOT), serum glutamate pyruvate transaminase (SGPT) from liver, kidney or spleen. Furthermore, we did not directly correlate whether altered peripheral organs are functionally associated with impairment of memory and executive function, as well as Aβ, tau, and other neurodegenerative biomarkers in the brains of these animal models. In addition, we did not investigate the early-stage changes in either of these organs of these mouse models of AD. In our previous studies, we have demonstrated the anti-amyloid, anti-inflammatory and synapto-protective effects of Cur in 5xFAD mice [24,25,26,27,28] and in animal models of Huntington’s disease [29]. Finally, our previous work supports the findings that Cur suppresses soluble tau oligomers in brain tissue and improved behavioral deficits in aged-human tau transgenic mouse models of AD [21]. We speculated that the CGR treatment could decrease pTau levels and decrease neuroinflammation in the brain tissue, similar to the peripheral organs, which was beyond the scope of the aims and objectives of the present study, so further work is needed. Taken together, our current findings of the peripheral organ damage are consistent with previous observations and are correlated with the impairment of higher brain functions in 3xTg mice.

## 5. Conclusions

Curcugreen is bioavailable to most of the peripheral organs and micromolar levels of it can protect against degenerative changes in liver, kidney, spleen and lungs of 3xTg and 5xFAD mice. Collectively, degenerative changes, as evidenced by splenomegaly, shrinkage of kidneys, huge emergence of monocytes and neutrophils in the spleen, and increased lung tissue damage suggest the development of an immunological dysfunction in 3xTg and 5xFAD mice. These pathological dysfunctions were ameliorated by CGR treatment. Collectively, these findings confirm that AD pathogenesis is not confined to the central nervous system, but it also involves metabolic dysfunction in the peripheral organs, and the treatments with CGR can prevent these changes.

## Figures and Tables

**Figure 1 antioxidants-10-00899-f001:**
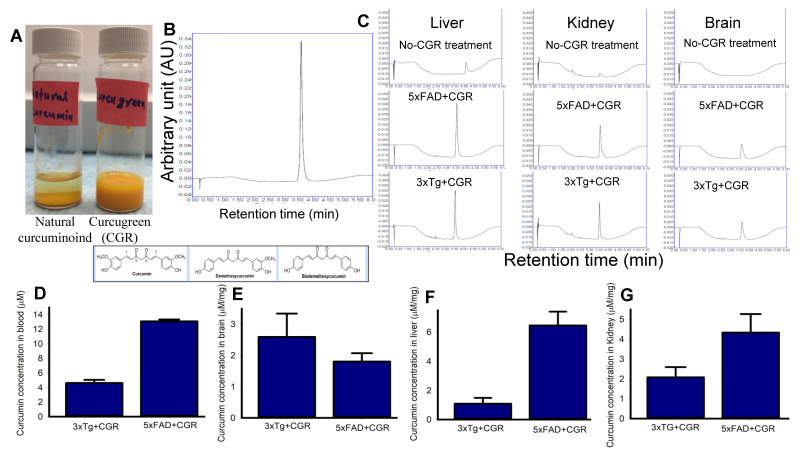
Curcumin levels in different tissues in 3xTg and 5xFAD mice after treatment with curcugreen. The 3xTg and 5xFAD mice were treated with curcugreen (100 mg/kg), every other day, orally, for 2 months and the blood, liver, kidney and brain tissues were collected, extracted and about 10 μL of filtrate was injected to UPLC for free Cur measurement. (**A**) Natural curcuminoids and CGR were dissolved in distilled water. Note that the CGR was dissolved completely in water, whereas natural curcuminoids appeared to precipitate. (**B**) Standard Cur (20 μM) peak and retention time when injected to UPLC. (**C**) Representative peak of Cur in CGR-treated 3xTg and 5xFAD mice from blood, liver, kidney and cerebellum. Note that no peak was observed in CGR-untreated mice tissues (upper panel), whereas CGR-treated mice tissues showed Cur peaks comparable to standard Cur peaks with similar retention time. (**D**–**G**) Amount of free Cur in blood (**D**), cerebellar tissue (**E**), liver (**F**) and kidney (**G**) of 3xTg and 5xFAD mice treated with CGR. The values were expressed and mean ± SEM from 4–5 mice in each group.

**Figure 2 antioxidants-10-00899-f002:**
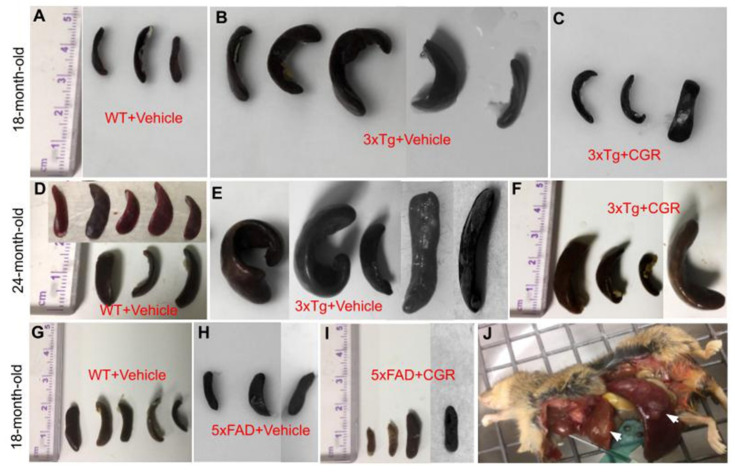
Splenomegaly was prevented by CGR treatment in 3xTg mice. Wild-type, 3xTg, 5xFAD mice were treated with CGR and the spleens were assessed after perfusion and fixation with 4% paraformaldehyde. (**A**–**F**): Spleens from WT+Vehicle (**A**), 3xTg+Vehicle (**B**) 3xTg treated with CGR, (**C**) 18-month-old WT+Vehicle (**D**), 3xTg+Vehicle (**E**), and 3xTg+CGR, and (**F**) 24-month-old 3xTg mice were measured. Note that the splenomegaly in 3xTg mice was significantly decreased in both 18- and 24-month-old groups after CGR treatment. (**G**–**I**) Spleens from WT+Vehicle (**G**), 5xFAD+Vehicle (**H**), and 5xFAD mice treated with CGR (**I**). Note that there was no significant change of spleen size in 5xFAD or 5xFAD+CGR mice in comparison to WT mice. (**J**) Representative image of a 24-month-old 3xTg mouse with splenomegaly and hepatomegaly (arrows).

**Figure 3 antioxidants-10-00899-f003:**
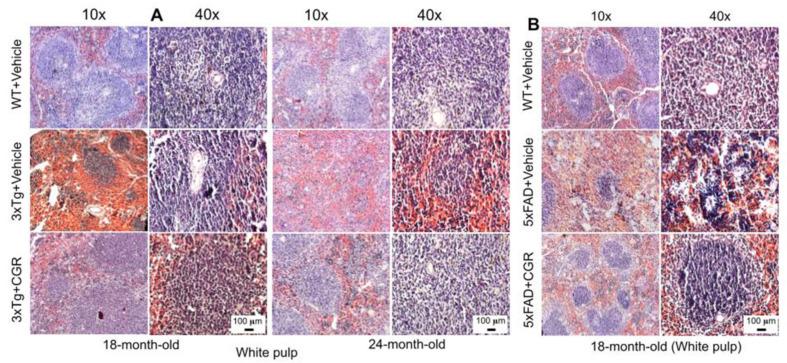
Splenic white-pulp cytoarchitecture was improved by CGR treatment in both 3xTg and 5xFAD mice. Wild-type, and 3xTg, 5xFAD mice treated with CGR (100 mg/kg) for 2 months and the mice were perfused with PBS and fixed with 4% paraformaldehyde and the liver was dissected out, embedded with paraffin and 5 μm thick sections were cut and stained with hematoxylin and eosin (H&E) and the images were taken by compound light microscope using either 10- or 40× objectives (total magnification = 100× or 400×). (**A**) Representative images of white pulp (splenic nodules, arrows) of WT+Vehicle, 3xTg+Vehicle and 3xTg+CGR mice from 18-month-old and 24-month-old mice. (**B**) Representative images of splenic white pulp from 18-month-old WT+Vehicle, 5xFAD+Vehicle and 5xFAD+CGR-treated mice. Note that 5xFAD+Vehicle treated mice showed a degeneration of white pulp and CGR treatment prevented these changes. A significant degeneration of white-pulp of spleen was observed in both 3xTg+Vehicle and 5xFAD+Vehicle-treated mice and treatment with CGR ameliorated these changes. Scale bar = 100 μm and is applicable to all other images.

**Figure 4 antioxidants-10-00899-f004:**
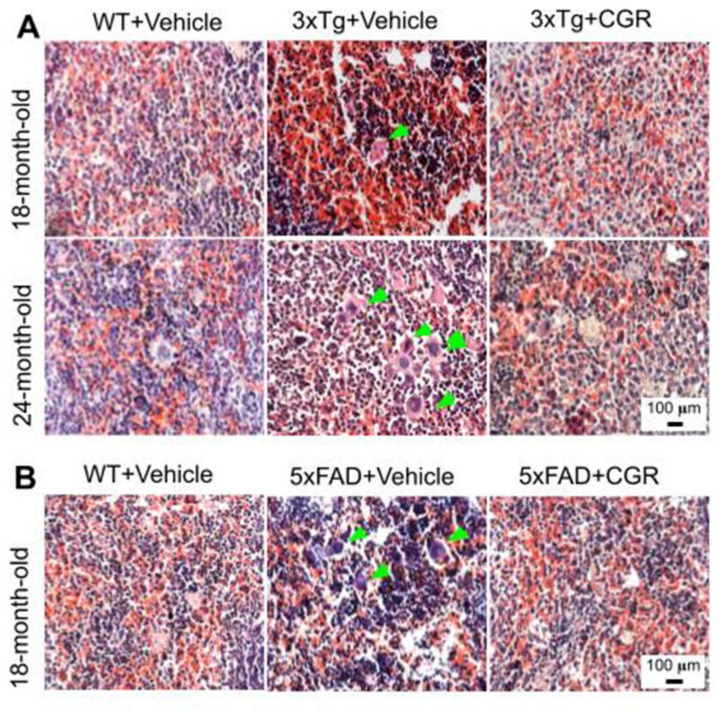
Splenic red-pulp cytoarchitecture was improved by CGR treatment in white-pulp of 5xFAD mice and red pulp of both 3xTg and 5xFAD mice. Wild-type, 3xTg, 5xFAD mice treated with CGR (100 mg/kg) for 2 months and the mice were perfused with PBS and fixed with 4% paraformaldehyde and the spleen was dissected out, embedded with paraffin and 5 μm thick section was cut and stained with hematoxylin and eosin stain and the images were taken by compound light microscope using either 10× or 40× objectives (total magnification = 100× or 400×). A: Representative images of splenic red pulp from 18- and 24-month-old 3xTg mice (**A**) and 18 months-old 5xFAD mice (**B**) after treatment with CGR. A significant degeneration of red-pulp, along with increased phagocytic cells (green arrows) in both these mice groups were observed, whereas CGR ameliorated these changes. Scale bar = 100 μm and is applicable to all other images.

**Figure 5 antioxidants-10-00899-f005:**
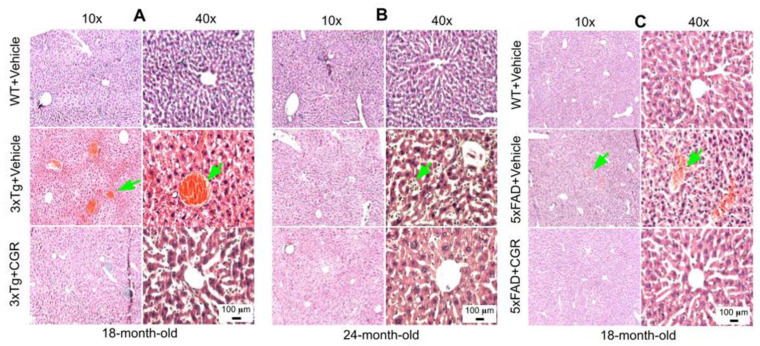
Liver morphology was improved by CGR treatment in both 3xTg and 5xFAD mice. Triple transgenic AD mice, 5xFAD, and age-matched WT mice were treated with CGR (100 mg/kg) for 2 months and the mice were perfused with PBS and fixed with 4% paraformaldehyde and the liver was dissected out, embedded with paraffin, and section at 5 μm, and stained with hematoxylin and eosin (H&E) stains and the images were taken by compound light microscope using either 10× or 40× objectives (total magnification = 100× or 400×). (**A**–**C**) Representative images of liver morphology from 3xTg mice from 18-month-old (**A**), 24-month-old (**B**) and 18-month-old 5xFAD mice (**C**) after treatment with CGR. A significant degeneration of liver central vein (green arrows) and lobules, as well as infiltrations were observed in both 3xTg and 5xFAD mice liver tissue and treatment with CGR ameliorated these changes. Scale bar = 50 μm and is applicable to all other images.

**Figure 6 antioxidants-10-00899-f006:**
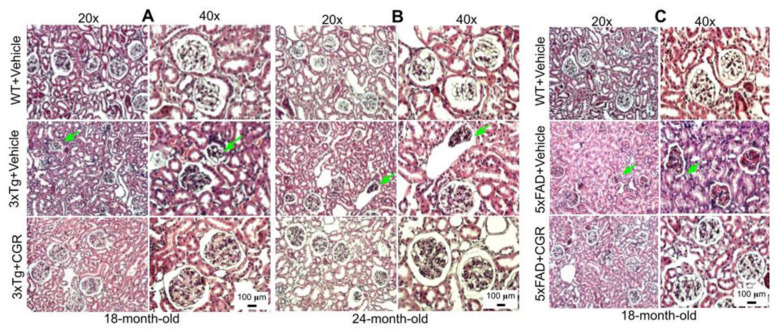
Kidney morphology was improved by CGR treatment in both 3xTg and 5xFAD mice. Wild-type, and 3xTg, 5xFAD mice were treated with CGR (100 mg/kg) for 2 months and the mice were perfused with PBS, fixed with 4% paraformaldehyde, and the kidneys were dissected out, embedded with paraffin, sectioned at 5 μm, and stained with hematoxylin and eosin (H&E) stains and the images were taken by compound light microscope using either 20× or 40× objectives (total magnification = 200× or 400×). (**A**–**C**) Representative images of Bowman’s capsules from the cortex of kidney of 3xTg mouse from 18-month-old (**A**), 24-month-old (**B**) and 18-month-old (**C**) 5xFAD mice after treatment with CGR. A significant shrinkage of renal tubules and degeneration of glomeruli were observed in both the 3xTg and 5xFAD mice (green arrows) and treatment with CGR ameliorated these changes. Scale bar = 50 μm and is applicable to all other images.

**Figure 7 antioxidants-10-00899-f007:**
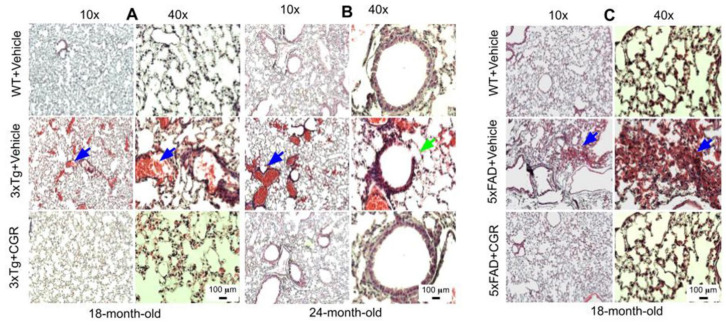
Lung morphology was improved by CGR treatment in both 3xTg and 5xFAD mice. Wild-type, 3xTg, and 5xFAD mice were treated with CGR (100 mg/kg) for 2 months and the mice were perfused with PBS and fixed with 4% paraformaldehyde and the lungs were dissected out, embedded with paraffin, sectioned at 5 μm and stained with hematoxylin and eosin (H&E) and the images were taken with a compound light microscope, using either 10- or 40-× objectives (total magnification = 100× or 400×). (**A**–**C**) Representative images of lung tissue from 18-month-old (**A**), 24-month-old (**B**) 3xTg mice and 18-month-old 5xFAD mice (**C**) after treatment with CGR. Significant damage of lung morphology, including blood infiltrations (blue arrows) and bronchial damage (green arrow), were observed in the lung tissue of both 3xTg and 5xFAD mice with treatments of CGR preventing these abnormalities. Scale bar = 50 μm and is applicable to all other images.

**Figure 8 antioxidants-10-00899-f008:**
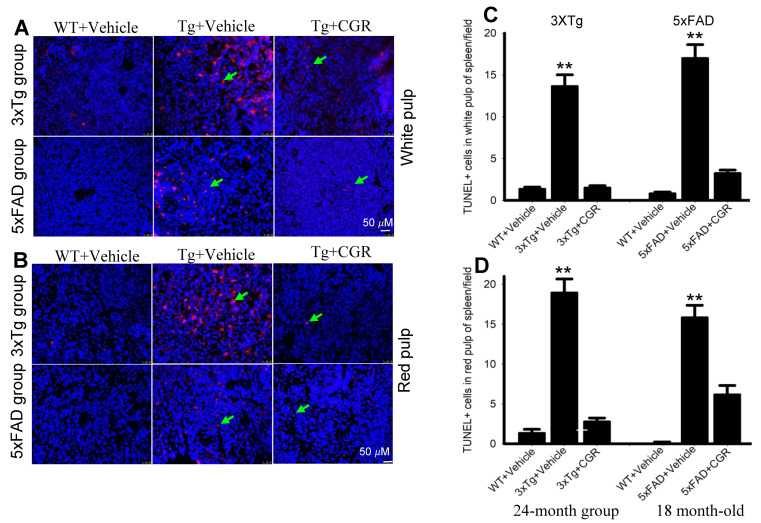
Number of TUNEL-positive cells was decreased in spleen by CGR treatment in both 3xTg and 5xFAD mice**.** After treatment with CGR (100 mg/kg) for 2 months, WT, 3xTg, and 5xFAD mice were perfused with PBS and fixed with 4% paraformaldehyde and in the spleens were dissected out, embedded with paraffin, and sectioned at 5 μm. The TUNEL assay was performed to detect DNA fragmented/apoptosis death of these tissues, followed by DAPI, as a counterstain, with images taken using a fluorescent microscope with appropriate excitation/emission filters at a total magnification of 400×. (**A**,**B**) Representative images of white pulp of spleen from 3xTg (24 months-old) and 5xFAD mice (18-month-old). (**C**,**D**) A significant increase in TUNEL-positive cells in white pulp (green arrows) of 3xTg+Vehicle-treated (**C**) and in 5xFAD+Vehicle-treated mice (**D**) in in comparison to age-matched WT+Vehicle-treated mice, whereas CGR treatment significantly reduced the number of TUNEL-positive cells. Red: TUNEL-positive cells, blue: DAPI (nuclei). Scale bar = 50 μm and is applicable to all other images. ** *p* < 0.01 in comparison to WT+Vehicle and 3xTg+CGR-treated mice. Blue= DAPI, Red= TUNEL-positive cells.

**Figure 9 antioxidants-10-00899-f009:**
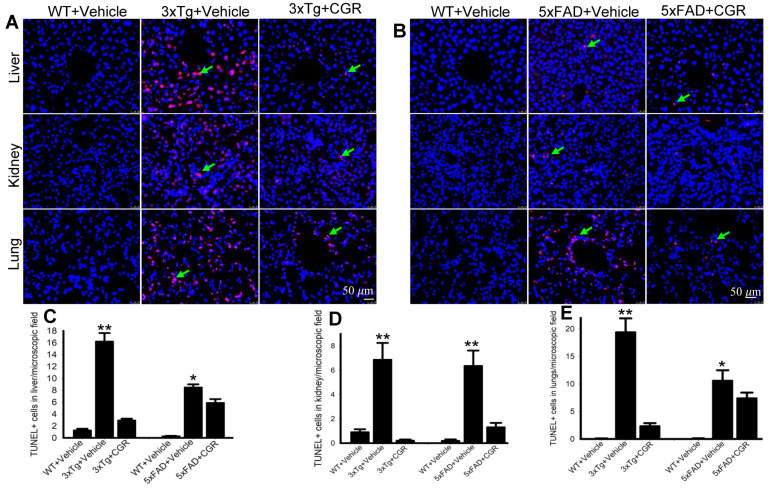
Number of TUNEL-positive cells was decreased in the liver, kidney and lungs by CGR treatment in both 3xTg and 5xFAD mice. After treatment with CGR (100 mg/kg) for 2 months, the WT, 3xTg, and 5xFAD mice were perfused with PBS and fixed with 4% paraformaldehyde and the liver, kidney and lung tissues were collected, embedded with paraffin and sectioned at 5 μm. The TUNEL assay was performed to detect DNA fragmented cells on these tissues, followed by DAPI, as a counterstain and the images were taken by fluorescent microscope (Leica, Germany) with appropriate excitation/emission filters using 40× objectives (total magnification = 400×). (**A**,**B**) Representative TUNEL staining photomicrographs of the liver, kidney and lungs from 24- month-old 3xTg mice (**A**) and 18-month-old 5xFAD mice (**B**) after treatment with CGR. (**C**–**E**) A significant increase in TUNEL-positive cells in the liver (**C**), kidney (**D**) and lungs (**E**) was observed in the 3xTg and 5xFAD mice, whereas CGR treatment significantly reduced the TUNEL-positive cells in these tissues. Red: TUNEL-positive cells, blue: DAPI (nuclei). Scale bar=50 μm and is applicable to all other images. * *p* < 0.05 and ** *p* < 0.01 in comparison to WT+Vehicle and 3xTg+CGR-treated or 5xFAD+CGR-treated mice. Blue = DAPI, Red = TUNEL-positive cells.

**Figure 10 antioxidants-10-00899-f010:**
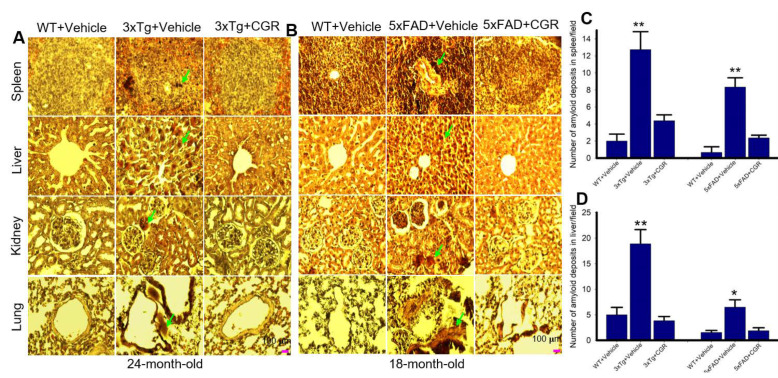
Amyloid deposition was reduced by CGR treatment in 3xTg and 5xFAD mice spleen, liver, kidney and lungs. After treatment with CGR (100 mg/kg) for 2 months, the WT, 3xTg and 5xFAD mice were perfused with PBS and fixed with 4% paraformaldehyde and the spleen were dissected out, embedded with paraffin and sectioned at 5 μm. The amyloid deposition was stained with Congo-red dye, followed using a counterstain with hematoxylin and the images were taken by compound light microscope with a 40× objective (total magnification = 400×). (**A**,**B**) Representative photomicrographs of the spleen, liver, kidney and lungs from 24-month-old 3xTg mice (**A**) and 18-month-old 5xFAD mice (**B**) after treatment with CGR. (**C**,**D**) Note that both the 3xTg and 5xFAD mice increased amyloid deposition (green arrows) in spleen (**C**), liver (**D**) and other tissues (graph not shown), whereas CGR treatment reduced the amyloid deposits in all these tissues. Scale bar = 100 μm and is applicable to all other images. * *p* < 0.05 and ** *p* < 0.01 in comparison to WT+Vehicle and Tg+CGR.

**Figure 11 antioxidants-10-00899-f011:**
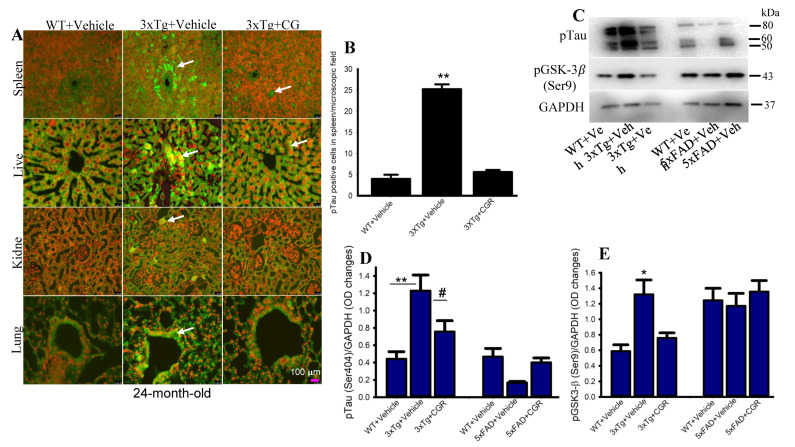
Phosphorylated tau immunoreactivity (pTau-IR) in peripheral organs was reduced by CGR treatment in 3xTg mice. Paraformaldehyde fixed, paraffin-embedded tissue sections at 5 μm from the WT, 3xTg and 3xTg+CGR mice were immunolabeled with pTau antibodies. The images were taken by a fluorescent microscope using 40× objectives (total magnification = 400×). (**A**) Representative photomicrographs of the spleen, liver, kidney and lungs from 24-month-old 3xTg mice after treatment with CGR. Note that the number of pTau-IR (arrows) were significantly increased in the spleen (**B**) and all other tissues (data not shown), whereas CGR treatment prevented the increase in the number of pTau-IR cells. (**C**–**E**) Western blot was performed from spleen tissue and labeled with pTau, pGSK-3β and GAPDH as a loading control. Note that pTau (**C**,**D**) and pGSK-3β (**C**,**E**) levels were significantly increased in 3xTg mice (not in 5xFAD mice) and CGR treatment significantly reduced those levels. Scale bar =100 μm and is applicable to all other images. * *p* < 0.05, ** *p* < 0.01 in comparison to WT and 3xTG+CGR; # *p* < 0.05 in comparison to 3xTg.

**Figure 12 antioxidants-10-00899-f012:**
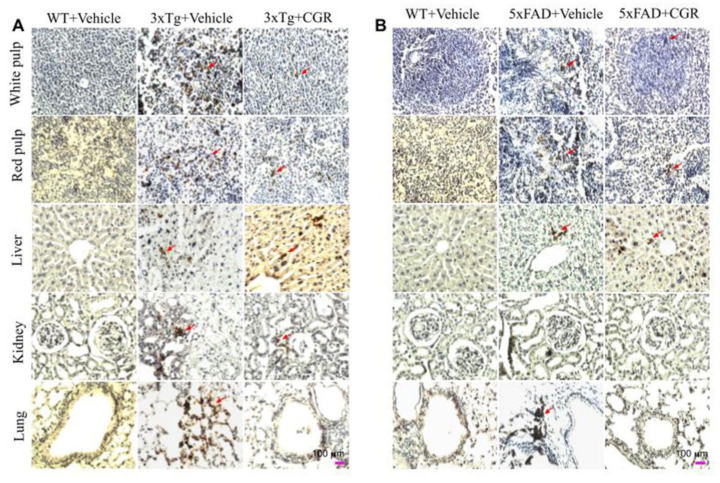
Cleaved caspase-3 immunoreactivity in the peripheral organs of 3xTg and 5xFAD mice after treatment with CGR. After paraformaldehyde fixation and paraffin-embedding, the spleen, liver, kidney and lung tissues from the WT, 3xTg (24-month-old) and 5xFAD (18-month-old) and their respective CGR-treated mice were immunolabeled with cleaved caspase-3 using avidin-biotin complex (ABC) method and the color was developed with diaminobenzedine (DAB). The images were taken by a compound light microscope using 40× objectives (total magnification = 40×). (**A**,**B**) Representative photomicrographs of the spleen, liver, kidney and lungs from 24-month-old 3xTg mice (**A**) and 18-month 5xFAD mice (**B**) after treatment with CGR. Note that the 3xTg and 5xFAD mice significantly increased the number of cleaved caspase-3 immunoreactivity (arrows) in all these tissues; the CGR treatment prevented these levels. Scale bar = 100 μm and is applicable to all other images.

**Figure 13 antioxidants-10-00899-f013:**
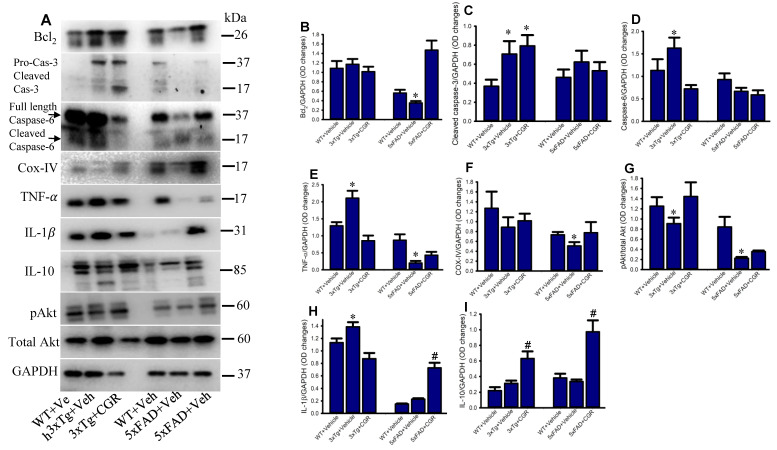
Curcugreen treatment reduced pTau, cell survival- and cell death and inflammatory markers in the spleen tissue from 3xTg and 5xFAD mice. Flash frozen spleen tissue from both 3xTg and 5xFAD mice were homogenized with an ice cold RIPA buffer with protease and phosphatase cocktail. Total protein was measured by BCA method and equal amounts of proteins were loaded and separated by sodium dodecyl sulphate gel electrophoresis (SD-PAGE), transferred to the PVDF membrane, after which they were probed with different primary antibodies. Signal was detected by a femto chemiluminescent kit and normalized with GAPDH as control. (**A**): Representative Western blot images of different data studied. (**B**–**I**): Graphs are representing the densitometric analysis of different parameters studied. Note that proinflammatory and cell death markers were increased, along with decreased anti-inflammatory markers in 3xTg and CGR treatment restored their levels. * *p* < 0.05 in comparison to WT and CGR-treated groups and # *p* < 0.05 in comparison to WT and 3xTg or 5xFAD mice.

## Data Availability

All the data analyzed for this manuscript are included. The analyzed raw data are available upon reasonable request to the corresponding author.

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
