# Peer review of "Curcugreen Treatment Prevented Splenomegaly and Other Peripheral Organ Abnormalities in 3xTg and 5xFAD Mouse Models of Alzheimer’s Disease"

_antioxidants, 2021, doi:10.3390/antiox10060899_

Round 1

Reviewer 1 Report

In this manuscript, the authors report that oral administration of Curcugreen (CGR) reduced degenerative changes in peripheral organs (Spleen, kidney, liver) and inflammatory responses in two models of AD mice (3xTg and 5X FAD).  While the results are of interest, there are concerns regarding the research design, data analysis and presentation, and data interpretation by the authors.

  • Details of bioavailability of CGR should be presented
  • Authors should emphasize the importance of this study as AD is associated with impaired CNS function including cognitive and memory loss. Verifying the effect of CGR on brain function would be more important.  
  • The images quality is good. However, the images for spleen morphology and spleen cytoarchitecture are same. Please present the correct images.
  • In fig 10. analysis for amyloid beta deposition between WT, 3XTg and 3xTg+CGR in both 18 and 24 month is required via image J.
  • In Fig 11. no details for C, D and E were shown in figure legend and discussed.
  • Baseline age related effects of naïve mice (18 months and 24 months old) should be established to comprehend the age-related effects. Therefore, brief details of young naïve adult mice would be appreciated.

Author Response

Comments and Suggestions for Authors

In this manuscript, the authors report that oral administration of Curcugreen (CGR) reduced degenerative changes in peripheral organs (Spleen, kidney, liver) and inflammatory responses in two models of AD mice (3xTg and 5X FAD).  While the results are of interest, there are concerns regarding the research design, data analysis and presentation, and data interpretation by the authors.

Comment: Details of bioavailability of CGR should be presented

Response: We have included a reference where the detailed bioavailability of CGR has been documented. Please see the reference no 31 in the revised manuscript.  

Comment: Authors should emphasize the importance of this study as AD is associated with impaired CNS function including cognitive and memory loss. Verifying the effect of CGR on brain function would be more important. 

Response: We completely agree with the reviewer's point. Our initial observation on peripheral organs abnormalities in 3xTg, especially the symptoms of splenomegaly prompted us to investigate the changes of the peripheral organs. We have started investigating the brain pathology of these mice after treatment with curcugreen and  in our ongoing project, we planned to compare and correlate the brain pathologies, including behavioral changes with these peripheral organ dysfunction in both 3xTg and 5xFAD mice after treatment with curcugreen. However, in recent research work, we have demonstrated the neuroprotective and behavioral improvement in 5xFAD mice after treatment with lipid formulation of curcumin [please see reference 28]. We have mentioned why we did not investigate the brain tissue in this study. Please see page 18, para 4, line 2-5, in the revised manuscript.

Comment: The image quality is good. However, the images for spleen morphology and spleen cytoarchitecture are the same. Please present the correct images.

Response: By mistake we uploaded the same image for Fig 3 and Fig 4. Thanks to the reviewer for catching this mistake. We have added the corrected images for Fig 3 and Fig 4 in the revised manuscript, please see at page no 8-9 in the revised manuscript.

Comment: In Fig 10. Analysis for amyloid beta deposition between WT, 3XTg and 3xTg+CGR in both 18 and 24 month is required via image J.

Response: We have included the morphometric graph for the amyloid plaque deposition in the revised manuscript. Please see revised Fig 10C-D, page 13 in the revised manuscript.

Comment: In Fig 11. no details for C, D and E were shown in figure legend and discussed.

Response: We apologize for the errors. In the revised manuscript at Fig 11, we have added the information for the details of the Fig 11A, B, C. Please see page 14.

Comment: Baseline age related effects of naïve mice (18 months and 24 months old) should be established to comprehend the age-related effects. Therefore, brief details of young naïve adult mice would be appreciated.

Response: Thanks for the reviewer's suggestion. In this study we have compared age-matched wild-type mice with 3xTg and 5xFAD mice. We understand that the comparison of the young naïve adult mice with these AD mouse models is important. For this, a new set of experiments need to run in order to explore these findings, and this is beyond the scope of this study.

Reviewer 2 Report

This work has investigated the overall histological changes and cell death mechanisms in spleen, liver, kidney, and lungs of the 3xTg and 5xFAD mouse models of AD after treatments with CGR or vehicle for 2 months period. The manuscript is well-planned and written although minor spelling mistakes shoud be corrected and Table 1 could be in supplementary or summarised. Strong results are showed in this study that allow to be published in Antioxidants. 

Author Response

Comments and Suggestions for Authors

Comment: This work has investigated the overall histological changes and cell death mechanisms in spleen, liver, kidney, and lungs of the 3xTg and 5xFAD mouse models of AD after treatments with CGR or vehicle for a 2 months period. The manuscript is well-planned and written although minor spelling mistakes should be corrected and Table 1 could be in supplementary or summarised. Strong results are shown in this study that allow them to be published in Antioxidants. 

Response: Thanks for the reviewer's  valuable comments and suggestions on our manuscript. We have replaced all the Tables as supplemental information. Please see supplemental materials.